# Function of Chloroplasts in Plant Stress Responses

**DOI:** 10.3390/ijms222413464

**Published:** 2021-12-15

**Authors:** Yun Song, Li Feng, Mohammed Abdul Muhsen Alyafei, Abdul Jaleel, Maozhi Ren

**Affiliations:** 1School of Life Sciences, Liaocheng University, Liaocheng 252000, China; songyun@lcu.edu.cn; 2Institute of Urban Agriculture, Chinese Academy of Agricultural Sciences, Chengdu 610213, China; fengli01@caas.cn; 3Zhengzhou Research Base, State Key Laboratory of Cotton Biology, Zhengzhou University, Zhengzhou 450001, China; 4Department of Integrative Agriculture, College of Agriculture and Veterinary Medicine, United Arab Emirates University, Al Ain P.O. Box 15551, United Arab Emirates; mohammed.s@uaeu.ac.ae (M.A.M.A.); abdul.jaleel@uaeu.ac.ae (A.J.)

**Keywords:** chloroplast, abiotic stress, biotic stress, plant regulatory response

## Abstract

The chloroplast has a central position in oxygenic photosynthesis and primary metabolism. In addition to these functions, the chloroplast has recently emerged as a pivotal regulator of plant responses to abiotic and biotic stress conditions. Chloroplasts have their own independent genomes and gene-expression machinery and synthesize phytohormones and a diverse range of secondary metabolites, a significant portion of which contribute the plant response to adverse conditions. Furthermore, chloroplasts communicate with the nucleus through retrograde signaling, for instance, reactive oxygen signaling. All of the above facilitate the chloroplast’s exquisite flexibility in responding to environmental stresses. In this review, we summarize recent findings on the involvement of chloroplasts in plant regulatory responses to various abiotic and biotic stresses including heat, chilling, salinity, drought, high light environmental stress conditions, and pathogen invasions. This review will enrich the better understanding of interactions between chloroplast and environmental stresses, and will lay the foundation for genetically enhancing plant-stress acclimatization.

## 1. Introduction

As sessile organisms, plants have to cope with various environmental conditions that are unfavorable or stressful for their growth and development. These adverse environmental conditions include abiotic stresses such as drought, heat, chilling/freezing, salinity, and nutrient deficiency as well as biotic stresses such as herbivore attack and pathogen infection [1,2,3,4]. These abiotic and biotic stresses limit crop yields and cause tremendous economic losses. These adverse effects are getting worse due to climate change, increasing population, and less arable land [1,2]. Plants have evolved a number of strategies to sense stress signals and adapt to adverse environments. Increasing evidence has revealed the essential role of the chloroplast, an organelle for photosynthesis in green plants, in plant stress response and stress adaption [3,4,5].

The chloroplast is a double-membrane plant endosymbiotic organelle where photosynthesis takes place [5,6,7,8]. Chloroplasts have retained their own genomes, and the chloroplast genome contains approximately 120 genes involved in chloroplast activities such as energy production and gene expression [9,10]. Chloroplasts produce energy through photosynthesis and oxygen-release processes, which sustain plant growth and crop yield. As such, chloroplasts are responsible for the biosynthesis of active compounds such as amino acids, phytohormones, nucleotides, vitamins, lipids, and secondary metabolites [9]. Furthermore, the chloroplast plays a vital role in plant acclimation to environmental stresses [3,4,6]. When plants are in adverse environmental conditions, chloroplasts sense these stresses and synthesize biologically active compounds and phytohormones, which protect plants from environmental stresses. In addition, chloroplasts communicate with the nucleus through plastid-to-nucleus retrograde signaling to acclimate to environmental stresses (Figure 1) [3,11,12].

Retrograde signaling pathways refer to the communications from the plastid to the nucleus [13]. The chloroplast functions as an environmental sensor. Fluctuations in the environment perturb chloroplast homeostasis, and the disturbance then drives the chloroplasts to communicate with the nucleus through retrograde signals. Eventually, plants remodel metabolism and gene expression to adapt to external stresses [7,11,12,13,14,15]. The retrograde signaling typically includes tetrapyrroles, phosphoadenosines, carotenoid oxidation products, isoprenoid precursors, carbohydrate metabolites, and reactive oxygen species (ROS) [16,17,18,19,20,21,22,23].

Under a series of unfavorable environmental conditions, chloroplasts generate the ROS retrograde signals including superoxide anion (O_2_^−^), hydrogen peroxide (H_2_O_2_), hydroxyl radical (OH▪), and singlet oxygen (^1^O_2_) [7,18,24,25,26,27]. The ROS in plants can severely threaten their health and viability. However, under stressful conditions, ROS function as a retrograde signal and modify the nuclear transcriptome to cope with these adverse stresses. Plants have evolved many strategies to maintain ROS dynamic equilibrium and normal photosynthetic efficiency, including complicated redox reaction chains and the ROS-scavenging system [18,26,27]. Singlet oxygen is generated at photosystem II (PSII) under high-light conditions through the excitation of ground-state triplet oxygen [7,11,24,28,29]. The singlet-oxygen-mediated transcriptional responses were first identified in the *flu Arabidopsis* mutant [30,31]. The functional FLU protein is a nuclear-encoded plastid protein that negatively regulates chlorophyll biosynthesis and consequently accumulates the strongly photosensitizing chlorophyll precursor protochlorophyllide. When dark-treated *flu* mutants are moved into the light, the mutants specifically accumulate singlet oxygen [32]. The singlet oxygen regulates a set of nuclear genes called singlet oxygen-responsive genes (SORGs), and most of these genes play a role in photosynthesis, carbon metabolism and plastid mRNA processing [7,33]. The activity of singlet oxygen signaling requires two nuclear-encoded proteins, Executor (Ex) 1 and 2, which are located in the thylakoid membrane of chloroplasts [31,32,34,35,36,37]. Furthermore, it has been reported that the plastid ATP-dependent zinc metalloprotease FtsH2 is involved in EX1/EX2 signaling [38,39]. Studies also find that β-cyclocitral is another singlet oxygen signaling pathway, which occurs independently of EX1 and EX2 [38,39,40]. Hydrogen peroxide acts as another retrograde signaling molecule. The specific role of hydrogen peroxide is identified through an RNAi line of thylakoid membrane-bound ascorbate peroxidase (tAPX) [41]. A number of putative components of the hydrogen peroxide signal transduction network were identified, including a series of transcription factors, mitogenactivated protein kinases (MAPKs) and miRNAs [7,42].

Methylerythritol cyclodiphosphate (MEcPP), an intermediate of the methylerythritol phosphate (MEP) pathway for plastid isoprenoid biosynthesis, functions as another retrograde signal to activate stress-responsive nuclear gene expression [23,43]. MEcPP was first identified by screening for mutants with elicited expression of hydroxyperoxide lyase (HPL). *HPL* is a nuclear stress-responsive gene and encodes a chloroplast enzyme, which plays a fundamental role in the defense-related molecules (such as the hormone jasmonic acid JA) synthesis [43]. The *constitutively expressing HPL* (*ceh1*) mutant was discovered. When exposed to high light or wounding stresses, plants accumulate MEcPP and regulate the expression of a series of nuclear genes [23,43,44,45]. Phosphonucleotide 3′-phosphoadenosine 5′-phosphate (PAP) is an important plastid metabolite and recently has emerged as a new retrograde signal for plant stress responses [16,46,47]. Chloroplast 3′ (2′), 5′-bisphosphate nucleotidase SAL1 can dephosphorylate PAP to adenosine monophosphate (AMP). When plants respond to drought and high light stresses, the activity of SAL1 is inhibited and PAP thus accumulates. PAP can regulate a series of nuclear stress-responsive genes [7]. Recent research has revealed the role of PAP in the drought stress response through regulating stomatal closure [46].

In this review, we will discuss the emerging function of chloroplasts in plant acclimation and adaption to environmental stresses including abiotic and biotic stresses. Our review focuses on findings from the past decade, and highlights the roles of chloroplast metabolites and chloroplast-derived signals in shaping how plants defend against harmful conditions. How chloroplast biology is affected by the changing environment is an emerging area of interest. Together, these studies highlight the important role of the chloroplast in plant adaptation to adverse environmental stresses. 

## 2. Chloroplast Response to Abiotic Stress

Advancing research has shown that chloroplasts play multifaceted roles in the plant response to various types of abiotic stress, including heat, chilling, salt, drought, and high light stresses. Here, we summarize the present state of knowledge on chloroplast responses to various abiotic stresses (Table 1).

### 2.1. Response to Heat Stress

Leaf photosynthesis is substantially affected by abnormal temperature stresses, including heat stress which is usually 10–15 °C above an optimum temperature for plant growth or chilling stress which occurs in the temperatures range 0–15 °C [27,48]. Chloroplasts play an essential role in activation of physiological adaptive processes to these adverse temperature stresses. 

It is reported that the chloroplast is sensitive to high-temperature stress during photosynthesis [48]. Research has revealed a close association between the chloroplast-related genes and high-temperature stress in the model plant rice. The expression of more than two hundred genes was upregulated in response to heat stress [49]. In a series of plant species, heat-induced leaf chlorosis has been observed [48]. After heat treatment, the activities of chlorophyll-degrading enzymes increased significantly, and the activity of key chlorophyll-synthesizing enzymes was unchanged [50]. Studies on genetic variations in hybrids of colonial (*Agrostis capillaris*) x creeping bentgrass (*Agrostis stolonifera*) indicated that the rapid breakdown of chlorophyll induced by heat stress was related to activation of genes encoding chlorophyllase and pheophytinase, and pheophytinase (PPH) activity [51]. The stay-green physiological traits were selected to evaluate the heat tolerance of plants and reveal the potential mechanism of heat damage associated with alterations in Chl metabolism and antioxidant and photosynthetic capacity. In creeping bentgrass species, adaptability to high temperature and stay-green genotypes was highly associated with chlorophyll metabolism [52]. Stay-green (SGR) genes encode magnesium dechelatase, and are involved in chlorophyll (Chl) degradation. Stay-green (SGR) homologs remove magnesium from Chl a, which is one of the most important components in the Chl degradation pathway in plants [53,54,55]. Under high-temperature conditions, accumulation of sugars in the peel was induced in bananas, and these sugars regulated Chl degradation through SGR proteins [56].

A series of research has shown that regulation of the activity of photosynthetic pathway enzyme Rubisco contributes to plant adaption to the heat stress. Rubisco activase (Rca1), a catalytic chaperone involved in modulating the Rubisco activity, plays a role in wheat response to heat stress [57]. The tomato (*Solanum lycopersicum*) chloroplast-targeted DnaJ protein (*Sl*CDJ2) is located in the thylakoids and stroma of the chloroplasts. When plants respond to heat stress, *Sl*CDJ2 protects Rubisco activity, and contributes to maintenance of CO_2_ assimilation capacity, which establishes a role for *Sl*CDJ2 in coping with heat stress [58]. The tomato (*Lycopersicon esculentum*) chloroplast-targeted DnaJ protein (*Le*CDJ1) also plays a role in plant response to high temperature [59]. 

Furthermore, the activity of PSI and PSII is severely affected by high-temperature conditions. In *A. thaliana*, heat stress can modulate the transcript accumulation of the plastid-encoded PSI and PSII genes such as the *psaA*, *psaB*, *psbA*, *psbD*, and *psbN*. Furthermore, the regulation is performed in part via the expression of HS-responsive nuclear genes for the plastid transcription machinery [60]. In wheat plants, the increased photosynthetic rate, improved ATPase activity in the thylakoid membrane, and enhanced efficiency of PSII photochemistry, which was achieved by overexpression of the ubiquitin/26S proteasome system *TaUb2*, contributed to plants coping with high-temperature stress [61]. When plants are exposed to transient heat waves, insufficient PSI photoprotection including the regulation of linear electron transport and the prevention of over-reduction, may affect the wheat photosynthetic capacity and make plants more susceptible to heat stress [62]. 

When plants were subjected to high-temperature stress, they generated large amounts of ROS and initiated related signaling events to survive the adverse environments [63]. In a recent study, researchers found that, under heat stress, the ROS levels contributed to mitigation of PSII photoinhibition in a coffee crop [64]. When plants respond to heat stress, the chloroplast heat-shock protein (Hsp) 21 becomes associated with the thylakoid membranes and plays a role in plant stress resistance [65]. 

### 2.2. Response to Low-Temperature Stress

In addition to high-temperature tolerance, chloroplasts are involved in plant cold-stress response. A series of studies have reported that chloroplasts can also perceive chilling stress signals, promote photosynthesis, and enhance plant resistance to adverse environment stress [27]. Previous and recent studies showed that low-temperature conditions could affect the abundance of various proteins involved in photosynthesis [27]. Through iTRAQ-based proteomic analysis between chilling-tolerant and chilling-sensitive rice lines, scientists revealed the dynamic response of chloroplast photosynthetic proteins under chilling conditions [66]. Systematic analysis of cold-stress response using transcriptome data was performed in rice, and the stay-green (SGR) proteins were identified as hub genes in this life process. Furthermore, SGR proteins were involved in the crosstalk between cold-stress responses and diurnal rhythmic patterns, providing new insights in understanding the plant environmental stress response against climate change [67].

Chilling can affect the structure, function, and development of chloroplasts [68]. Scientists revealed that chilling can induce structural changes during chloroplast biogenesis in cucumber cotyledons [69]. When exposed to low temperatures, plant chloroplasts changed the content of unsaturated fatty acids in chloroplast membranes to increase plant tolerance to the adverse temperatures [70]. When plants adapted to chilling stresses, the activities of chloroplast dark-reaction-related enzyme were also regulated. For instance, the reductive activity of two key enzymes involved in the Calvin cycle, fructose-1,6-diphosphatase (FBPase) and isoheptanone-1,7-diphosphatase (SBPase) was significantly reduced [71]. Low-temperature stress hinders chloroplast development and plant photosynthesis. The rice (*Oryza sativa*) RNA-binding protein DUA1 is required for RNA editing of the rps8-182 site, and plays a vital role in chloroplast development under low-temperature conditions [72]. The chloroplast gene *psbA* encodes the key D1 reaction center protein of PSII. Recently, the tomato (*Solanum lycopersicum*) WHIRLY1 (*Sl*WHY1) was found to be induced by chilling conditions and could upregulate the transcription level of *psbA* through directly binding to the upstream region of its promoter (the sequence “GTTACCCT”). The increased D1 abundance enhanced plant resistance to photoinhibition caused by chilling stresses [73]. Scientists also found that *Sl*WHY1 could increase the expression level of RbcS1, a member of the Rubisco small-subunit (RBCS) multigene family, and help plants maintain high Rubisco content under low-temperature conditions [74]. Despite the role of chloroplast-targeted DnaJ protein in plant response to heat stress, the function of tomato (*Lycopersicon esculentum*) chloroplast-targeted DnaJ protein (*Le*CDJ1) under chilling stress was also investigated. *Le*CDJ1 showed essential functions in maintaining PSII activity under low-temperature stress [75]. 

The ROS function has an important retrograde signal in chloroplasts under a series of unfavorable environmental conditions. Chilling stress can regulate the redox state of chloroplasts. When plants respond to chilling stress, the photosynthetic electron transport chain in chloroplasts transfers excess electrons to O_2_ and causes O_2_^-^ increase in bermudagrass [76]. In *A. thaliana*, regulation of chloroplast-to-nucleus ROS signaling is a strategy to promote plants acclimation to cold stress [77]. Researchers revealed that melatonin increases the chilling tolerance of cucumber seedlings through regulating the ROS balance [78]. Scientists have also found that, under chilling stress, exogenous applications of acetylsalicylic acid can enhance the chloroplast antioxidant system activity and thus improve the tolerance of plants under low temperatures [79]. 

It is reported that the chloroplast RNA-binding protein RBD1 plays a critical role in plant chilling tolerance. RBD1 directly binds to 23S rRNA, and influences 23S rRNA processing in *Arabidopsis*, which suggests the importance of chloroplast function especially protein translation in coping with chilling stresses [80]. 

### 2.3. Response to Salt Stress

The chloroplast also regulates plant response to salt stress. In soybean, salt stress can decrease the chloroplast Rubisco amount, and Rubisco-containing body (RCBs) formation is responsible for this life process [81]. Sugar beet is a highly salt-tolerant plant and is a model for studying salt acclimation in crops. When sugar beets were exposed to salt stress, they could adjust their cellular redox and reactive oxygen species (ROS) network to improve their salinity tolerance [82]. Under salinity stress conditions, plants modulate protein transport into chloroplasts to enhance tolerance to stresses through upregulating the transcription level of the ubiquitin E3 ligase suppressor of PPI1 locus 1 (SP1). SP1 induces degradation of translocon at the outer envelope membrane of chloroplasts (TOC), inhibits photosynthesis, and decreases ROS formation [83,84]. Recently, it has been reported that the chloroplast Hsp70 chaperone protein and Clp protease systems can protect plants from the environmental stress conditions such as salinity-triggered oxidative stress. The Hsp70 chaperone systems refold proteins after stress, while Clp protease degrades the misfolded and aggregated proteins [85]. In cabbage (*Brassica rapa*), the chloroplast-targeted DEAD-box RNA helicase *BrRH22* expression was induced by high salinity or drought stress and by ABA treatment. *Br*RH22 had the RNA chaperone function and affected chloroplast gene translation, eventually positively regulating plant response to abiotic stress [86]. To some extent, the stay-green (SGR) phenotype protects the plants against abiotic stress conditions. In rice, overexpression transgenic lines of RNA-binding bacterial chaperones encoding genes *CspA* and *CspB* displayed a distinct stay-green (SGR) phenotype, higher transcript levels of SGR, and enhanced multiple stress tolerance. This research indicated that bacterial chaperone proteins are capable of imparting SGR phenotype and salt- and drought-stress tolerance alongside grain improvement [87].

### 2.4. Response to High-Light Stress

Extreme changes in light intensities in the environment are also adverse environmental stresses for plants. When *A. thaliana* seedlings respond to drought and high light stresses, a SAL1-PAP retrograde pathway in chloroplast can regulate the expression of a subset of high-light and drought-inducible nuclear genes, and enhance their acclimation to these stresses [16]. By analysis of the changes in chloroplast transcript accumulation and translation in leaves of tobacco (*Nicotiana tabacum*) seedlings after transferring from moderate light to physiological low or high light, scientists found that the transcription level of only a single chloroplast gene *psbA* were significantly changed, suggesting the limited responsiveness of chloroplast gene expression during adaption to high light in tobacco [88]. When grape leaves responded to the combined stress of high light and high temperature, the cyclic electron flow around PSI was stimulated and played an important role in PSI and PSII photoprotection [89]. During plants’ acclimation to fluctuating light, chloroplasts send information to the nucleus. In *Nicotiana benthamiana* epidermal cells, high light increased chloroplast H_2_O_2_ production, and the produced H_2_O_2_ was transferred directly from chloroplasts to nucleus to control nuclear gene expression [90]. The transcriptome of *A. thaliana* cell suspension culture under high-light conditions was analyzed. Key components of the ROS signaling transduction pathway and transcription factors that regulate ROS scavenging were upregulated during early transcriptional responses to high-light stress [29]. Scientists found that the thylakoid membrane protease FtSH in *Chlamydomonas reinhardtii* play a vital role in the quality control of thylakoid membrane proteins and in the response to light stress [91]. 

### 2.5. Response to Drought Stress

Chloroplasts also play a pivotal role in plant acclimatization to drought stresses [3,7,92]. As mentioned above, when plants respond to drought stress, the SAL1-PAP chloroplast retrograde pathway regulates nuclear stress-responsive genes and stomatal closure to improve plant drought tolerance [16,46]. Drought stress can induce the expression of chloroplast-targeted DEAD-box RNA helicase *Br*RH22 in cabbage, which regulates plant response to abiotic stress [86]. The rice gene *OsCTR1* encodes the RING Ub E3 ligase, and its encoded protein is localized in the chloroplasts. *Os*CTR1 interacted with two chloroplast-localized proteins *Os*CP12 and *Os*RP1. Heterogeneous overexpression of *Os*CTR1 transgenic *Arabidopsis* plants was ABA-sensitive and showed improved tolerance against severe water deficits, indicating the involvement of *Os*CTR1 in plant response to water-deficit stress [93]. Researchers performed RNA-Seq analysis using roots from 4-week-old rice seedlings grown in soil that had been subjected to drought conditions for 2–3 d [94]. They found that rice phytochrome B (*Os*PhyB) repressed the activity of ascorbate peroxidase and catalyzed mediating reactive-oxygen-species (ROS) processing machinery required for drought tolerance, suggesting the potential significance of *Os*PhyB for manipulating drought tolerance in rice [94]. Maize mesophyll lipoxygenase *Zm*LOX6, a vegetative storage protein (VSP), was targeted to chloroplasts by a novel N-terminal transit peptide. Overexpression of *Zm*LOX6 improved plant drought tolerance. Furthermore, additional storage of nitrogen by *Zm*LOX6 as a VSP in maize leaves ameliorated the adverse effect of drought on grain yield [95]. The *Salsola laricifolia* NADP–malic enzyme (NADP–ME) *Sa*NADP–ME4 was found to be located in the chloroplasts and was highly expressed in leaves. *Sa*NADP–ME4 alleviated the chlorophyll content decrease and PSII photochemical efficiency, and enhanced reactive oxygen species scavenging capability. *Arabidopsis* overexpressing *Sa*NADP–ME4 showed increased drought-stress resistance [96]. Under moderate drought conditions, the B-box (BBX) gene *AtBBX21* overexpression can reduce chloroplast electron transport capacity to increase photosynthesis and improve water-use efficiency in potato (*Solanum tuberosum*) plants. The function of BBX21 was achieved through the interplay with the ABA signaling networks [97]. *Sorghum landraces* have evolved in drought-prone regions. Recently scientists investigated the genetics of their adaptation through quantitative and population genomics. They found that the stay-green loci played a broad role in the drought adaptation of sorghum [98].

## 3. Chloroplast Response to Biotic Stress

Under the attack of biotic agents including arthropods, fungi, and bacterial and viral pathogens, damage is caused to the chloroplasts and plant photosynthesis. Recently, the chloroplasts have emerged as pivotal factors to coordinate plant defense against biotic agents (Table 1) [4,15,99,100,101].

The transcriptome data from eight different plant species exposed to biotic damage were analyzed. As previously reported, the plant-hormone-related genes including jasmonic-acid-, salicylic-acid-, and ethylene-responsive genes were upregulated [100]. More importantly, the photosynthesis-related gene expression was universally downregulated, and the downregulation was an adaptive response to biotic attack, suggesting a role of chloroplasts in coordinating plant biotic defense [102]. Researchers revealed that activation of defense by pathogen-associated molecular patterns (PAMPs) leads to a rapid decrease in nonphotochemical quenching (NPQ), and NPQ also influences plant responses of PAMP-triggered immunity. These results indicate that the regulation of chloroplast NPQ is an intrinsic component of the plant’s defense program [103]. In the presence of pathogen infection, chloroplasts send out dynamic tubular extensions called stromules to communicate with nucleus [104,105]. Their communication is accompanied by the accumulation of chloroplast-localized defense protein NRIP1 and H_2_O_2_ in the nucleus, indicating the importance of this communication for successful innate immune responses [104]. Recently, the underlying biological mechanism for the life process in which chloroplasts cluster around nuclear and extended stromules was further uncovered. In *Nicotiana benthamiana*, scientists found that this response can be triggered by activation of pattern-triggered immunity, ETI, or infection with viruses or bacteria and this response is non-cell-autonomous [106]. 

PP2C is one of the four major classes of the serine/threonine-specific protein phosphatase family. In *A. thaliana*, the PP2C genes (*AtPP2C62* and *AtPP2C26*) double-mutant plants showed enhancement of plant immunity and resistance to *Xanthomonas campestris* pv. *campestris* infection. Furthermore, the *At*PP2C62 and *At*PP2C26 protein localized to the chloroplast, and catalyzed the dephosphorylation of the photosynthesis-related protein chaperonin-60. This study implied the occurrence of crosstalk between photosynthesis and the plant defense system [107]. The candidates influencing plant life processes related to the formation and maintenance of stromules were also investigated in *Nicotiana benthamiana* using *Xanthomonas campestris* pv. *vesicatoria* type III-secreted effectors. Overexpression of E3 ubiquitin ligase XopL eliminated the formation of stromules and the relocation of chloroplasts to the nucleus, and microtubules played a vital role in stromule extension and dynamics [108].

Wheat-stripe rust is a devastating disease caused by the fungus *Puccinia striiformis* f. sp *tritici* (*Pst*). The *WHEAT KINASE START1* (*WKS1*) gene encodes a serine/threonine kinase and confers partial resistance to *Pst*. The wheat (*Triticum aestivum*) WKS1.1 is targeted to the chloroplast and phosphorylates the thylakoid-associated ascorbate peroxidase (tAPX) to reduce its ability to detoxify peroxides [109]. Furthermore, researchers also found WKS1 interacted with and phosphorylated an extrinsic member of PSII PsbO, which resulted to reduced photosynthesis and conferred *Pst* resistance [110]. The chloroplastic protein thylakoid formation1 (THF1) plays a vital role in maintaining chloroplast homeostasis. THF1 is a negative regulator of cell death, and can be destabilized by the nucleotide-binding/Leu-rich repeat (NB-LRR) protein N’, revealing novel molecular mechanisms linking light and chloroplasts to the induction of cell death in plant immunity [111]. Recently, the function of the rice thylakoid membrane-bound ascorbate peroxidase *Os*APX8 in rice tolerance to bacterial blight *Xanthomonas oryzae* pv. *Oryzae* was investigated. The overexpression of *OsAPX8* could enhance the rice tolerance to bacterial pathogen by regulating H_2_O_2_ accumulation [112]. In rice, the phosphorylation of light-harvesting complex II protein LHCB5 enhances broad-spectrum resistance of rice to the blast fungus *Magnaporthe oryzae* through modulating ROS accumulation in the chloroplasts. This study uncovered an immunity mechanism mediated by phosphorylation of light-harvesting complex II [113].

In *Arabidopsis* response to nonhost *Pseudomonas syringae* pathogens, plants lacking the functional redox detoxification system NADPH-dependent thioredoxin reductase C (NTRC) showed enhanced disease susceptibility of nonhost pathogens and elevated JA-mediated signaling pathways. These results suggested NTRC plays a key role in plant immunity by regulating chloroplast-generated reactive oxygen species [114]. The *Arabidopsis* pathogen-responsive mitogen-activated protein kinases (MPKs) MPK3 and MPK6 play important roles in promoting plant defense against pathogens. When plants respond to *P. syringae* pv tomato, the MPK3/MPK6 protein manipulated plant photosynthetic activities and promoted ROS accumulation in chloroplasts, and eventually contributed to the robustness of ETI pathway [115].

The genome-wide expression changes were examined in *Arabidopsis* leaves following challenge with *P. syringae* pv tomato DC3000. Scientists found that suppression of chloroplast-associated genes represented a rapid MAMP-triggered defense response, further suggesting the key role of chloroplasts in plant defense [116]. The N-terminal regions of *P. syringae* pv. tomato DC3000 type-III effector HopK1 represented a chloroplast transit peptide, indicating that the type-III effectors targeted the chloroplasts and the targets within this organelle were important for immunity [117]. The bacterial pathogen *P. syringae* pv tomato DC3000 cysteine protease effector HopN1 localized to chloroplasts, degraded the oxygen evolving complex of photosystem II PsbQ, and inhibited PSII activity in chloroplast preparations, indicating that PsbQ was a target for pathogen suppression and contributed to plant immunity responses [118]. Upon *P. syringae* pv tomato DC3000 challenge, virus-induced gene-silenced *SlALC* and *Arabidopsis thf1* mutants exhibited accelerated lesion formation, and *Sl*ALC1 chloroplast localization was affected by coronatine [119].

Recently, the role of chloroplastic ROS in plant defense against a typical necrotrophic fungus *Botrytis cinerea* was investigated. A series of experiments in transgenic *Nicotiana tabacum* (tobacco) lines expressing a plastid-targeted cyanobacterial flavodoxin (pfld lines) indicated that chloroplast-generated ROS play a major role in lesion development during *Botrytis* infection. Modulation of chloroplastic ROS levels by the overexpression of a heterologous antioxidant protein can also protect plants against the canonical necrotrophic fungus *B. cinerea* [120]. 

*Ralstonia solanacearum*, a soilborne plant pathogen, invades host plant roots from the soil. This plant pathogen injects type III effector proteins called Rips (*Ralstonia*-injected proteins) into plant cells. Scientists found that *R. solanacearum* RipAL localized to chloroplasts and targeted chloroplast lipids. *R. solanacearum* RipAL induced JA production and suppressed SA-signaling-mediated defense response in plant cells [121]. In wheat, the necrotrophic pathogens *Stagonospora nodorum* and *Pyrenophora tritici-repentis* could produce effector ToxA, and wheat sensitivity to ToxA was uniquely governed by the *Tsn1* gene. The expression level of *Tsn1* was tightly regulated by the circadian clock and light, indicating that Tsn1-ToxA interactions are associated with photosynthesis pathways [122]. The Irish potato famine pathogen *Phytophthora infestans* secretes the effector protein AVRvnt1, and this protein can be recognized by the nucleotide-binding leucine-rich repeat (NLR) protein Rpi-vnt1.1. In the dark, Rpi-vnt1.1-mediated resistance was compromised, indicating that pathogens manipulated the chloroplast function and resulted in a light-dependent immune response [123]. *Ss*ITL is an effector protein of the necrotrophic phytopathogen *Sclerotinia sclerotiorum*, and suppresses host immunity at the early stage of infection. Recently, scientists found that *Ss*ITL interacted with a chloroplast-localized calcium-sensing receptor CAS, interfered with the plant salicylic acid (SA) signaling pathway, and facilitated the infection by *S. sclerotiorum* [124].

**Table 1 ijms-22-13464-t001:** Summary of known chloroplast biological processes involved in plant stress response.

Components	Species	Process/Stimulus	Molecular Function	Reference
Heat stress				
SGR	*Musa acuminata*	Chlorophyll degradation	Stay-green protein	[56]
Rca1	*Triticum aestivum*	Modulate the activity of Rubisco	A catalytic chaperone	[57]
CDJ2	*Solanum lycopersicum*, *Lycopersicon esculentum*	Protect Rubisco activity	Chloroplast-targeted DnaJ protein	[58,59]
Ub2	*Triticum aestivum*	Improve antioxidant capacity	Ubiquitin/26S proteasome system	[61]
Hsp21	*Arabidopsis thaliana*	Associate with the thylakoid membranes	Small heat-shock protein chaperone	[65]
Chilling stress				
FBPase, SBPase	*Zea mays* L.	Compensate for decreases in photosynthetic capacity	Enzymes involved in the Calvin cycle	[71]
DUA1	*Oryza sativa*	Regulate chloroplast development	RNA-binding protein	[72]
WHY1	*Solanum lycopersicum*	Upregulate the PSII key D1 reaction center protein encoding gene *psbA*; maintain high Rubisco content	WHIRLY proteins—the plant-specific DNA-binding proteins	[73,74]
CDJ1	*Lycopersicon esculentum*	Maintain PSII activity	Chloroplast-targeted DnaJ protein	[75]
RBD1	*Arabidopsis thaliana*	Regulate chloroplast protein translation by influencing 23S rRNA processing	Chloroplast RNA-binding protein	[80]
Salt stress				
SP1	*Arabidopsis thaliana*	Induce the degradation of translocon at the outer envelope membrane of chloroplasts (TOC)	Uiquitin E3 ligase suppressor of PPI1 locus 1	[83,84]
Hsp70, Clp protease	*Arabidopsis thaliana*	Protect plants from salinity-triggered oxidative stress	Hsp70—chaperone; Clp—a protease system	[85]
RH22	*Brassica rapa*	Affect chloroplast gene translation	Cloroplast-targeted DEAD-box RNA helicase	[86]
CspA, CspB	*Oryza sativa*	Impart SGR phenotype	RNA-binding bacterial chaperones	[87]
High-light stress				
SAL1-PAP	*Arabidopsis thaliana*	Regulate stress-inducible nuclear genes	Components of retrograde pathway	[16]
FtSH	*Chlamydomonas reinhardtii*	Control the quality of thylakoid membrane proteins	Thylakoid membrane protease	[91]
Drought stress				
SAL1-PAP	*Arabidopsis thaliana*	Regulate the stress-inducible nuclear genes	Components of retrograde pathway	[16]
RH22	*Brassica rapa*	Affect chloroplast genes’ translation	Chloroplast-targeted DEAD-box RNA helicase	[86]
CTR1	*Oryza sativa*	Interact with two chloroplast-localized proteins, OsCP12 and OsRP1	RING Ub E3 ligase	[93]
PhyB	*Oryza sativa*	Repress the activity of ascorbate peroxidase	Phytochrome B	[94]
LOX6	*Zea mays*	Additional storage of nitrogen	Mesophyll lipoxygenase in chloroplast	[95]
NADP-ME4	*Salsola laricifolia*	Alleviate chlorophyll content decrease and PSII photochemical efficiency	NADP-malic enzyme	[96]
BBX21	*Solanum tuberosum*	Reduce chloroplast electron transport capacity	B-box (BBX) protein	[97]
Biotic stress				
PP2C62, PP2C26	*Arabidopsis thaliana*	Catalyze the dephosphorylation of the photosynthesis-related protein, chaperonin-60	Components of the serine/threonine-specific protein phosphatase family	[107]
XopL	*Nicotiana benthamiana*	Eliminate stromules formation and chloroplast relocation	E3 ubiquitin ligase	[108]
WKS1	*Triticum aestivum*	Phosphorylate thylakoid-associated ascorbate peroxidase (tAPX) and detoxify peroxides; phosphorylate an extrinsic member of photosystem II (PSII) PsbO	A serine/threonine kinase	[109,110]
THF1	*Nicotiana benthamiana*	Maintain chloroplast homeostasis	Chloroplastic protein thylakoid formation1	[111]
APX8	*Oryza sativa*	Regulate H_2_O_2_ accumulation	Thylakoid membrane-bound ascorbate peroxidase	[112]
LHCB5	*Oryza sativa*	Regulate ROS accumulation	Light-harvesting complex II protein	[113]
NTRC	*Arabidopsis thaliana*	Modulate chloroplast-generated ROS	Redox detoxification system NADPH-dependent thioredoxin reductase C	[114]
MPK3, MPK6	*Arabidopsis thaliana*	Manipulate plant photosynthetic activities and promote ROS accumulation	Mitogen-activated protein kinase	[115]
PsbQ	*Nicotiana benthamiana, Arabidopsis thaliana*	A target for pathogen suppression and contributes to plant immunity responses	The oxygen evolving complex of photosystem II	[118]
ALC	*Solanum lycopersicum*, *Arabidopsis thaliana*	Regulate disease-associated necrotic cell death	Chloroplast genes with altered responses to coronatine	[119]
RipAL	*Ralstonia solanacearum*	Localized to chloroplasts and targeted chloroplast lipids	Type III effector proteins	[121]
Tsn1	*Triticum aestivum*	Interact with effector protein ToxA	A unique wheat disease resistance-like gene, regulated by the circadian clock and light	[122]
Rpi-vnt1.1	*Nicotiana benthamiana*	Recognize the effector protein AVRvnt1, and mediate a light-dependent immune response	A nucleotide-binding leucine-rich repeat (NLR) protein	[123]
CAS	*Arabidopsis thaliana*	Recognize the effector protein, and interfere with the salicylic acid (SA) signaling pathway	A chloroplast-localized calcium-sensing receptor	[124]

## 4. Conclusions and Future Perspectives

Adverse environmental stresses greatly influence plant growth and development. Photosynthesis, one of the most important anabolic processes in plants, is severely inhibited by adverse conditions.

In crop species, the reduced photosynthesis rate under adverse environmental conditions eventually causes severe yield and economic losses. It is well known that the chloroplast is a unique organelle in green plant cells for hosting photosynthesis. Recent studies, as summarized in this review, have indicated that chloroplasts also act as sensors of external environment stresses and communicate via a series of retrograde signals to the nucleus, and thus they play an essential role in plant response to abiotic and biotic stresses. 

In recent decades, extraordinary progress has been made in the field of elucidating the role of the chloroplast in the plant’s response to abiotic and biotic stresses. To facilitate plant survival under adverse environmental stresses, chloroplasts regulate their structure, gene expression, protein remodeling, and the related metabolic pathways including chloroplast-sourced oxylipins, hormones, hydrogen peroxide, and singlet oxygen. However, future challenges in this field are complex and multiple. First, whether the responses in chloroplasts are inevitable consequences of adverse environmental-stress induction and whether these responses have certain roles in inducing plant tolerance to stress damage. Second, how environmental stresses trigger chloroplast response. Third, how chloroplast responses are transferred out of the chloroplast and induce multiple retrograde signaling pathways to communicate with the nucleus. Finally, whether the present retrograde signaling pathways from the chloroplast to the nucleus under stress conditions are mainly related to reactive oxygen species, and whether other retrograde signaling pathways are involved in the chloroplast’s response to stresses needs further investigation. Dissection of these questions will help understanding of the precise function of chloroplasts in the plant stress response and lay the foundation for genetically improving plant-stress resilience. 

## Figures and Tables

**Figure 1 ijms-22-13464-f001:**
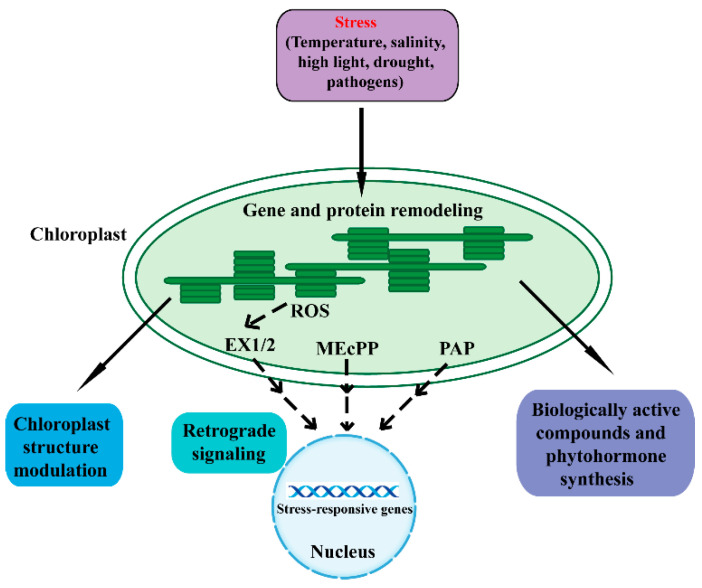
An overview of the various mechanisms in chloroplast response to adverse stresses. Adverse environmental stresses cause perturbations and generate signals in chloroplasts that regulate chloroplast gene expression and protein remodeling. A series of cellular activities are then triggered to restore chloroplast homeostasis. Adverse conditions can affect the structure, function, and development of chloroplasts. Chloroplasts synthesize biologically active compounds and phytohormones to acclimate to stresses. Moreover, the chloroplast is able to communicate its status to the nucleus through retrograde signaling to regulate nuclear stress-responsive genes. The SAL1/PAP, MEcPP, and ROS pathways act as important components of the chloroplast retrograde signaling pathway. Dashed lines indicate postulated regulation. ROS, reactive oxygen species; EX1/2, executor 1/2; MEcPP, methylerythritol cyclodiphosphate; PAP, phosphonucleotide 3′-phosphoadenosine 5′-phosphate.

## Data Availability

Not applicable.

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
