# Peer review of "Function of Chloroplasts in Plant Stress Responses"

_ijms, 2021, doi:10.3390/ijms222413464_

Round 1
Reviewer 1 Report
It is a well written document that reflects the current state of the art, presents good scientific quality, making it suitable for a review article.
It covers the various topics related to the subject, which are well supported by recent and adequate bibliography.
However some more bibliographical references in the area of abiotic stress can be included.
Minor spelling errors were found throughout the document that should be corrected. Some examples:
line 124, 2.1. Reaponse to heat stress
line 169, 2.2. Reaponse to low temperature stress
line 221, Uner salinity stress conditions
Overall it is a good scientific contribution and should be accepted for publication after minor revision.
Author Response
Thanks for your constructive suggestions. As suggested, we have added some more bibliographical references in the area of abiotic stress. Furthermore, we have examined and corrected the minor spelling errors throughout the manuscript. Now, the overall quality of the manuscript has been improved.
Reviewer 2 Report
1) References can be added (e.,g line 35, line 42, line 90 etc.).
2) Figure 1 is not referred to anywhere in the text.
3) The MS needs revision for typos throughout the text. Some examples:
* line 82 replace “are” in “were”
*line 91 replace “highlight” in “high light”
*line 105 replace “thees” in “those”
* line 147 CO2
*line 204 O2
*line 201 replace “as” in “has”
4) Line 216 “osmotic stress” – salt stress is both osmotic and ionic. I think that salt stress is the correct phrase here
5) This review aim to address a very important topic in plant biology which is the emerging role of chloroplast in plant stress tolerance. As is, few major topics are not covered at all. One is the VERY important and common abiotic stress of water deficiency and the other is the accumulating evidence for the stay green (SGR) trait and genes and stress induced chloroplast degradation in regulating plant stress tolerance. I recommend addressing those topics as well to some extent.
Author Response
Thanks for your constructive suggestions. We have addressed the concerns and suggestions from you in the revised manuscript. We have carefully edited and corrected the entire manuscript for grammar and accuracy with expert from a native English speaker. The overall language of this paper is substantially improved. Following apart is the point-to-point response to your comments.
1) References can be added (e.,g line 35, line 42, line 90 etc.).
Response: Thanks for the comments. We have added the related references in the revised manuscript.
2) Figure 1 is not referred to anywhere in the text.
Response: Thanks. Figure 1 is referred to line 48 in our original text and line 52 in the revised version.
3) The MS needs revision for typos throughout the text. Some examples:
* line 82 replace “are” in “were”
*line 91 replace “highlight” in “high light”
*line 105 replace “thees” in “those”
* line 147 CO2
*line 204 O2
*line 201 replace “as” in “has”
Response: Thanks for your suggestions. We have gone through the manuscript and corrected all the spelling mistakes.
4) Line 216 “osmotic stress” – salt stress is both osmotic and ionic. I think that salt stress is the correct phrase here
Response: Done as suggested. We have changed the “osmotic stress” to “salt stress”.
5) This review aim to address a very important topic in plant biology which is the emerging role of chloroplast in plant stress tolerance. As is, few major topics are not covered at all. One is the VERY important and common abiotic stress of water deficiency and the other is the accumulating evidence for the stay green (SGR) trait and genes and stress induced chloroplast degradation in regulating plant stress tolerance. I recommend addressing those topics as well to some extent.
Response: Thanks for your suggestions. In the revised version, we have addressed the researches about the role of chloroplasts in plant response to drought stress and the stay green trait in regulating plant stress tolerance, and the overall quality of the manuscript has been improved.
Round 2
Reviewer 2 Report
The authors have fully addressed all the comments and the MS has been appropriately revised.